

**An innovative eddy-covariance system with vortex intake for**
**measuring carbon dioxide and water fluxes of ecosystems**
**Jingyong Ma[1,2], Tianshan Zha[1,2], Xin Jia[1,2], Steve Sargent[3], Rex Burgon[3], Charles**
**P.-A. Bourque[4], Xinhua Zhou[3], Peng Liu[1,2], Yujie Bai[1,2], Yajuan Wu[1,2]**
[1] School of Soil and Water Conservation, Beijing Forestry University, Beijing, China
[2] Beijing Engineering Research Center of Soil and Water Conservation, Beijing Forestry
University, Beijing, China
[3] Campbell Scientific, Inc., Logan UT, USA
[4] Faculty of Forestry and Environmental Management, 28 Dineen Drive, University of
New Brunswick, Fredericton, New Brunswick, Canada
*Correspondence to*: Tianshan Zha (tianshanzha@bjfu.edu.cn)





**Abstract.** Closed-path eddy-covariance (EC) systems are used to monitor exchanges
of carbon dioxide ($CO_2$) and water vapor ($H_2O$) between the atmosphere and biosphere.
Traditional EC intake systems are equipped with in-line filters to prevent airborne dust
particulates from contaminating the optical windows of the sample cell which degrades
measurements. In order to preserve fast-frequency response, the in-line filter should be
small, but small filters plug quickly and require frequent replacement. This paper
reports the test results of a field-performance of an innovative EC system (EC155,
Campbell Scientific, Inc.) with a prototype vortex intake replacing the in-line filter of
a traditional EC system. The vortex intake design is based on fluid dynamics theory. An
air sample is drawn into the vortex chamber, where it spins in a vortex flow. The initially
homogenous flow is separated when particle momentum forces heavier particles to the
periphery of the chamber, leaving a much cleaner air stream at the center. Clean air (75%
of total flow) is drawn from the center of the vortex chamber, through a tube to the
sample cell with optical windows. The remaining 25% of the flow carries the heavier
dust particles away in a separate bypass tube. An EC155-system measured $CO_2$ and
$H_2O$ fluxes in two urban forest ecosystems in the megalopolis of Beijing, China. These
sites present a challenge for EC measurements because of the generally poor air quality
with high concentrations of suspended particulate. The closed-path EC system with
vortex intake significantly reduced maintenance requirements by preserving optical
signal strength and sample cell pressure within acceptable ranges for much longer
periods. The system with vortex intake also maintained excellent high-frequency
response. For example at the Badaling site, percentage system downtime due to plugged


filters was reduced from 26% with traditional in-line filters to 0% with the prototype
vortex intake. The use of vortex intake could extent the geographical applicability of
the EC technique in ecology and allow investigators to acquire more accurate and
continuous measurements of $CO_2$ and $H_2O$ fluxes in a wider range of ecosystems.




















# 1 Introduction

Eddy-covariance (EC) technology provides an opportunity to evaluate the fluxes of

energy, momentum, water vapor, carbon dioxide, and other scalars between the earth's

surface and the turbulent atmosphere overhead (Montgomery, 1948; Baldocchi, 2003;

Aubinet et al., 2016). The technology has been widely used in ecosystem studies

worldwide, including in forests, grasslands, agricultural lands, and wetlands (e.g., Zha

et al., 2010; Mitchell et al., 2015; Shoemaker et al., 2015; Wang et al., 2015). However,

the technology's use in many urban greenspace ecosystems has been challenged by

polluted air that contaminates the optical windows of the gas analyzer. Optical signal

strength is reduced and gas concentration measurements degrade as dust and debris are

deposited on the optical windows of the analyzer, including in both open and closed

path system. Closed path systems with traditional in-line filters can help keep the

analyzer's windows free of debris for a longer time. However, in environments with

extremely dirty air, in-line filters plug quickly (in just a few days) and as a result, require

frequent replacement (Bressi et al., 2012; Yu et al., 2013; Hasheminassab et al., 2014;

Villalobos et al., 2015). Dirty sample air can also contaminate other parts of the EC

system, leading to underestimated fluxes and data gaps (Jia et al., 2013; Xie et al., 2015).

In extreme cases, the system can stop working properly.

With expanding urbanization, urban greenspaces are expanding commensurately

(Pataki et al., 2006). Urban greenspaces have been playing a progressively more

important role in the study of ecosystem carbon balances worldwide (Mchale et al.,



2007). To address the challenges associated with urban settings, an improved EC-system capable to operate in polluted urban environments is needed to monitor carbon dynamics in urban areas and to evaluate greenspace-ecosystem response to environmental change (e.g., Pataki et al., 2006; Xie et al., 2015).

The traditional approach for maintaining good trace-gas concentration measurements in a closed-path EC system is to use an in-line filter to clean sampled air. The in-line filter in the original EC155 design is based on a sintered stainless steel disk 1/16-inch thick x 1 inch diameter of either 20 or 40 µm porosity, mounted in a rubber rain cap. In practice, the porosity is chosen to maximize the maintenance-free interval at specific sites. Fine-pore filters keep the analyzer windows clean for a longer time, but plug more quickly. The gas analyzer sample cell windows must be cleaned when the optical signal strength diminishes to 80%, and the filter must be replaced when the pressure drop exceeds 7 kPa. Ideally, the filter pore size is chosen such that the windows become dirty at the same interval that the filter clogs, requiring just one maintenance visit to the site. The inline filter is a functional solution that can filter dirty air sufficiently to maintain good gas concentration measurements. However, the maintenance labor can be significant, and either a clogged filter or dirty windows can disrupt measurements until the analyzer can be maintained. This maintenance can be required frequently in conditions with high particulate matter in the ambient air.

To avoid the frequent replacement of filters in EC systems deployed in urban



environments, an advanced EC system with vortex intake (United States Patent No.
9,217,692)has been recently developed by Campbell Scientific, Inc. Vortex intake
eliminates the need for an in-line filter upstream of the gas analyzer.

This study introduces vortex-intake sampling and demonstrates its field performance
with *in situ* measurements collected in two urban greenspace areas within the
megalopolis of Beijing, China. The goals for the new design were to: (1) minimize
system maintenance; (2) reduce system downtime due to plugged filters; and (3)
maintain high-frequency response. The objective of this field test was to compare the
performance of a prototype vortex-intake sampling system with that of a traditional
system fitted with an in-line filter.
**2 Materials and methods**
**2.1 Site description and data collection period**
The study site is located in Beijing Olympic Forest Park (40.02° N, 116.38° E, 51 m
above mean sea level, AMSL) and Badaling Tree Farm (40.37° N, 115.94° E, 535 m
AMSL), Beijing, China. There is generally poor air quality in Beijing, with high
concentrations of suspended particulate in the atmosphere (Fig. 1) and hazy conditions,
at times with visibility < 10 km. Haze is a common problem during the winter and
spring because of (i) home heating with coal and other non-renewable energy sources,
(ii) congested traffic, (iii) industrial activity, (iv) stable synoptic conditions, and (v)
surrounding mountainous topography (e.g., Yang et al., 2015; Zheng et al., 2015; Zhang
et al., 2016).




The Olympic Forest Park is the largest urban forest park in Asia, with an area of 680 ha
and vegetation coverage of about 90%. The site is an ecological conservation and
recovering area. The site is dominated by *Pinus tabulaeformis L*. Other species include
*Platycladus orientalis*, *Sophora japonica L*, *Fraxinus chinensis*, and *Glingo biloba*,
with an understory of *Iris tectorum* and *Dianthus chinensis*. All trees were tagged and
identified by species, with trees with diameter at breast height (DBH) > 3 cm being
assessed annually. Stand density was 210 trees ha$^{-1}$, with a mean tree height of 7.7 m
and a mean DBH of 20 cm. Cover ratio of trees to shrubs was about 7:3. The shrubs
were *Prunus davidiana, Amygdalus triloba, Swida alba*, and *Syzygium aromaticum*,
with a mean height of 2.8 m (Xie et al., 2015).

The Badaling Tree Farm is about 60-km away from downtown core of Beijing. Local
terrain is generally flat and uniform. The study site is composed of *Acer truncatum*,
*Koelreuteria paniculata, Fraxinus bungeana, Ailanthus altissima*, and *Pinus*
*tabuliformis*. Stand density was 975 trees ha$^{-1}$, with a mean tree height of 4 m and a
mean DBH of 4.7 cm. The study site has a sparse herbaceous cover layer with no well-
defined understory canopy (Jia et al., 2013).

Data presented in this study cover the field deployments of two EC systems; one at each
of the two sites. Both systems were deployed in 2011 with the original inline filter
intakes. Data with this inline filter design were collected from January 2011 to July



2014 at the Olympic Park, and from January 2011 to September 2014 at Badaling Farm.
Both EC systems were switched to the vortex design in 2014. Data with vortex intakes
were acquired from July 2014 to December 2015 at Olympic Park and from September
2014 to December 2015 at the Badaling Farm.

## 160   2.2 Instrument description

An EC155 (model EC155, Campbell Scientific, Inc. Logan, UT, USA) is an *in situ*,
closed-path, mid infrared absorption gas analyzer (IRGA) that measures molar mixing
ratios of $CO_2$ and $H_2O$ at high frequency. The original EC155 analyzer includes a heated
intake tube, inline filter, and rain cap (Fig. 2a). The modified EC155 system includes a
prototype vortex chamber and rain cap in place of the original filter and rain cap (Fig.
2b).

The vortex intake is a small, light-weight device with no moving parts, and requires no
chemicals to clean the sample air. Its simple design makes it essentially maintenance
free. The vortex assembly (Fig. 2b) consists of a rain cap and inlet nozzle, a vortex
chamber, and two outlet ports. Schematics of both systems are shown in Fig. 3. Unlike
a filter, the vortex intake design is based on fluid and particle dynamics. Sampled air
enters the vortex chamber through a tangent port to induce rotational flow. Entrained
dust-particle motion is governed by centrifugal (inertia), aerodynamic drag, and
chamber wall-impact forces. The high rotational speed of the vortex flow provides the
larger heavier (relative to air) dust particles with greater centrifugal force, keeping them
close to the chamber wall and leaving the air in the center of the vortex free of dust.





Clean sample air flows from the vortex center through a tube to the EC155 sample cell.
The dust particles along with the air close to the wall of the vortex chamber are pulled
out through the bypass tube. This dirty air passes through a relatively large filter that
lasts a long time without plugging. The filter protects a flow-control orifice that
balances the flow split ¾ to sample, ¼ to bypass. The two flows rejoin downstream of
the analyzer then go to the single low power vacuum pump.

For flux determination (covariances), high-frequency wind velocities are needed. Wind
velocities are acquired with a fast-response 3-dimensional sonic anemometer (CSAT3A;
Campbell Scientific, Inc. Logan, UT, USA). At Olympic Forest Park, a 12-m-tall tower
is surrounded by uniform forest cover with a homogeneous fetch of about 600 m in all
directions. The EC instruments were mounted on a tower of 11.5-m height from ground.
At Badaling Tree Farm, the EC instruments were mounted on a tower of 11.7-m height
from ground. All flux-related data were collected at 10 Hz and processed using Fluxnet-
supported methodologies described by Aubinet et al. (1999).
**2.3 Field tests**
High-frequency fluctuation amplitude-reduction is one of the systematic errors in EC
measurements, and is especially important when trace gas concentrations are measured
with a closed-path infrared gas analyzer (Aubinet et al., 1999). The frequency response
of a system is defined as a ratio of its output to its input as a function of the signal
frequency. In an ideal system, the value would be unity across all frequencies. In a real
system, fluctuations in $CO_2$ or $H_2O$ tend to be damped at higher frequencies due to





adding a rain cap, filter, and intake tubing to a gas sampling system (Aubinet et al.,
2016). The frequency response quantifies the loss of high-frequency information and
tests the ability of the frequency response corrections to adequately account for this loss.

With this in mind, the frequency response of a closed-path EC system with vortex intake
is compared to one with an in-line filtering setup. Laboratory test of the frequency
response using the method of Sargent (2012) showed that closed-path eddy covariance
systems with vortex intake perform very well. *In situ* field measurement of system
frequency response is challenging because the true signal inputs (scalar variables) are
not known *a priori*.

*In situ* system-frequency response can be evaluated by comparing the cospectra of the
vertical component of wind with fluctuations of sonic temperature (WTs, where W
denotes the vertical component of wind and Ts the sonic temperature) to the cospectra
of the vertical component of wind with fluctuations of $CO_2$ (WC, where C denotes the
mixing ratio of $CO_2$) and $H_2O$ (WH, where H denotes $H_2O$ mixing ratio). Analysis of
cospectrum are based on high-frequency data at 10 Hz obtained from the Olympic
Forest Park over a one-hour period (12:00-13:00 Beijing Standard Time) averaged daily
for the month of April, 2014, for the in-line filter-based EC measurements, and August,
2015, for the vortex intake-based measurements. A fast Fourier transform was applied
for each variables' time series consisting of about 36,000 data points.



Decreases in sample-cell differential pressure and $CO_2$ optical signal strength indicate
when the in-line filter and optical windows need to be replaced and cleaned. Clogged
filters can induce substantial pressure drops (Aubinet et al., 2016). Generally, the
pressure drop in the original intake assembly is approximately 2.5 kPa at 7 LPM flow
without filter. The filter adds approximately 1 kPa pressure drop when it is clean. This
pressure drop will increase as the filter clogs. The filter should be replaced before the
differential pressure reaches +/- 7 kPa. Additionally, the windows of the analyzer should
be cleaned when the optical signal strength of $CO_2$ drops below 80% of the original
value.

## 3 Results and discussion

### 3.1 Frequency response

Due to damping of high-frequency signals in closed-path systems, gas cospectra
commonly exhibit reduced response at high frequencies causing flux loss (Leuning and
King, 1992; Burba et al., 2010). Brach and Lee (Brach et al., 1981; Lee et al., 2004)
found that the cospectrum of vertical wind velocity with sonic temperature (WTs) is
often very close to the ideal cospectrum. In field experiments, WTs is often used as a
standard to evaluate whether there is a high-frequency loss for other measured scalars.
To examine the effect of vortex intake, spectral analysis was applied to the
measurements collected *in situ*. Ensemble cospectra of vertical wind velocity with $CO_2$
(WC) and $H_2O$ vapor (WH) were compared to those for the sonic temperature (WTs)
for both the in-line filter and vortex intake-based systems (Fig. 4a, b). The normalized
cospectra for both systems were consistent at all frequencies, with no significant



difference($P > 0.05$), thus the EC155 sampling system with vortex intake did not result
in obvious $CO_2$ and $H_2O$ flux losses. Additionally, although the cospectra of WC and
WH for both systems showed a sharp attenuation when the frequencies > 0.1 Hz,
damping of high-frequency signals was within the range reported for closed-path
systems (Yasuda et al., 2001). The EC155 system with vortex intake can provide
sufficient high-frequency response to warrant automating high-frequency spectral
corrections during post-processing.
**3.2 Differential pressure and optical signal strength of $CO_2$**
Air quality at Olympic Park was mostly poor during the measurement periods, being
worse in winter than in summer. To verify performance of the vortex intake, we chose
periods of high and low incidence of haze to compare the differential pressure and
optical signal strength of $CO_2$. Fig. 5 shows time series of three months for each case.
Decrease in differential pressure caused by clogging of the in-line filter (Fig. 5a and c)
quickly exceeded the range of the pressure sensor (+/- 7 kPa), generating invalid data
until the filter could be replaced. The differential pressure with the vortex intake was
much more stable than with the in-line filter. Over a period of three months, the
differential pressure with vortex intake exceeded the pressure sensor range once during
very hazy conditions (Fig 5b). The optical signal strength of $CO_2$ with the vortex intake
remained above 90%, indicating the optical windows remained free of debris for a
substantially longer time than with the in-line filter.

At the Badaling Farm site, we also chose periods of high and low incidence of haze to





compare differential pressure and optical signal strength of $CO_2$. As shown by the
differential pressure in Fig. 6a and c, the in-line filter clogged multiple times in a period
of two months resulting in large data losses. Pressure drop with the vortex intake (Fig.
6b and d) was typically about 3 kPa, remaining within the working range for the entire
observation period (two months). The optical signal strength of $CO_2$ with vortex intake
was higher and more stable than that of the system with an in-line filter.
**3.3 Field maintenance**
In order to further verify the performance of the EC155 system with vortex intake, field
maintenance records from the two sites were compared. These maintenance records
included the number of maintenance services, downtime due to clogged intake filters,
and percentage downtime, defined as a ratio of downtime due to clogged intake to the
actual testing-period duration × 100. Also included is the minimum and average time
period for the intake filter to clog.

A summary of field maintenance at Olympic Park and Badaling Farm is shown in Table
1. The vortex design reduced the number of maintenance services from 15 to 4 at
Olympic Park, and from 7 to 0 at Badaling Farm. The percentage downtime at Badaling
was reduced from 26% with the original in-line filter intake to 0% with the vortex intake.
At Olympic Park the percentage downtime was 31% for the inline filter and 5% for
vortex intake. The percentage downtime reflects not just the number of times the filter
was clogged, but also how soon the filter was replaced after it clogged. The minimum
and average clog times, shown in the last two columns of Table 1, highlight the reduced



maintenance requirement of the vortex intake design. At Olympic Park, the in-line
filters clogged in as little as one day, with an average clog period of just six days. The
minimum maintenance interval for the vortex intake was 21 days, with an average of
46 days. At Badaling Farm, in-line filters clogged in as little as 9 days with an average
of 20 days, while the vortex intake design required no maintenance for an entire period
of 122 days.

Independent of high or low haze cover, the maintenance required for the vortex intake
EC system was markedly reduced, thus decreasing overall downtime substantially.
Similar field tests for the vortex intake EC system have been performed in the USA and
Canada (Brown et al., 2015; Somers and Sargent, 2015), where it ran well over six
months with no maintenance required on either the vortex sampling system or sample
cell windows. Overall, the vortex intake can improve long-term monitoring of $CO_2$ and
$H_2O$ fluxes in conditions of high particulate concentration.

## 4 Conclusions

The vortex intake significantly reduced maintenance requirements and down-time for
a closed-path eddy-covariance system compared to the original in-line filter design.
Vortex intake kept the sample cell windows cleaner, preserving the optical signal
strength of $CO_2$ longer. Its installation also avoided the need for an in-line filter in the
sample path, sustaining an acceptable sample cell differential pressure over a much
longer period. There was no significant attenuation of high frequencies, compared to
the in-line filter-based system. Vortex intake helped to overcome shortcomings





associated with the traditional in-line filter-based systems in extremely polluted
conditions. The vortex intake design extents the geographical application of the EC
technique in ecology and allows investigators to acquire more accurate and continuous
measurements of $CO_2$ and $H_2O$ fluxes in a wider range of ecosystems.

*Acknowledgments.* The research was supported by grants from National Natural
Science Foundation of China (NSFC; 31670710, 31670708, 31270755, 31361130340),
and the Fundamental Research Funds for the Central Universities (Proj. No. 2015ZCQ-
SB-02). The U.S.–China Carbon Consortium (USCCC) supported this work via helpful
discussions and the exchange of ideas. The authors acknowledge Karen Wolfe for
technical writing and editing support, Campbell Scientific, Inc. Logan, UT, USA, and
Wenqing Hu and Xiaojie Zhen, BTS, Beijing, China. We are grateful to Cai Ren and
Cai Zhang for their assistance with the field measurements and instrumentation
maintenance. We also would like to thank anonymous reviewers and the editors for their
constructive comments on this manuscript.











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



## Tables and Figures

**Table 1.** Summary of field maintenance notes for the EC155 with vortex intake compared to the

EC155 with an in-line filter at Olympic Park (op) and Badaling Farm (bd), Beijing, China; the time

range includes periods of both high and low incidence of haze. Clog period is the number of days

for the differential pressure to reach +/- 7 kPa after installing a fresh filter.

| Site | Begin data | End data | Intake | Time range (days) | Number of maintenance services | Downtime from clogged intake (days) | Downtime percent | Minimum clog period (days) | Average clog period (days) |
|---|---|---|---|---|---|---|---|---|---|
| **op** | 1/11/13 | 31/1/14 | Inline filter | 184 | 15 | 57 | 31% | 1 | 6 |
| | 1/8/13 | 31/10/13 | | | | | | | |
| | 1/11/14 | 31/1/15 | Vortex | 184 | 4 | 9 | 5% | 21 | 46 |
| | 1/8/15 | 31/10/15 | | | | | | | |
| **bd** | 1/11/12 | 31/12/12 | Inline filter | 122 | 7 | 32 | 26% | 9 | 20 |
| | 1/9/12 | 31/10/12 | | | | | | | |
| | 1/11/14 | 31/12/14 | Vortex | 122 | 0 | 0 | 0% | >122 | >122 |
| | 1/9/15 | 31/10/15 | | | | | | | |



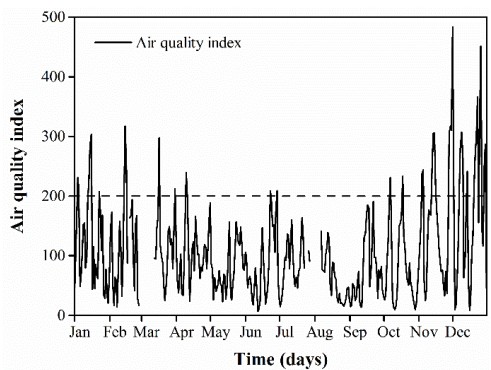


**Figure 1.** Daily mean air quality index during 2015 in Beijing, China. Air quality index is stratified

into six categories: 0-50 for low, 51-100 for low to mild, 101-150 for mild, 151-200 for moderate,

201-300 for severe, and > 300 for serious air pollution levels.





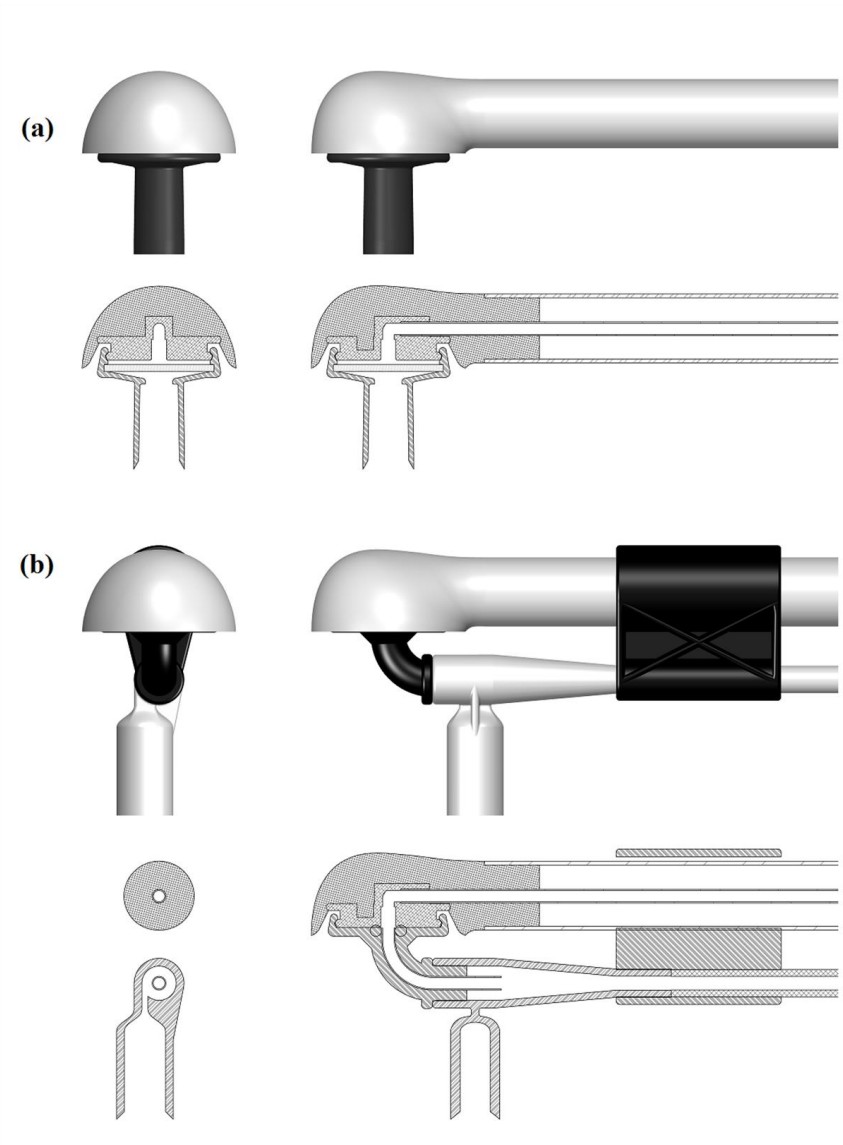


**Figure 2.** EC155 sample intakes: (a) original in-line filter and (b) prototype vortex intake cleaner

(source: Campbell Scientific Inc., Logan UT, USA).





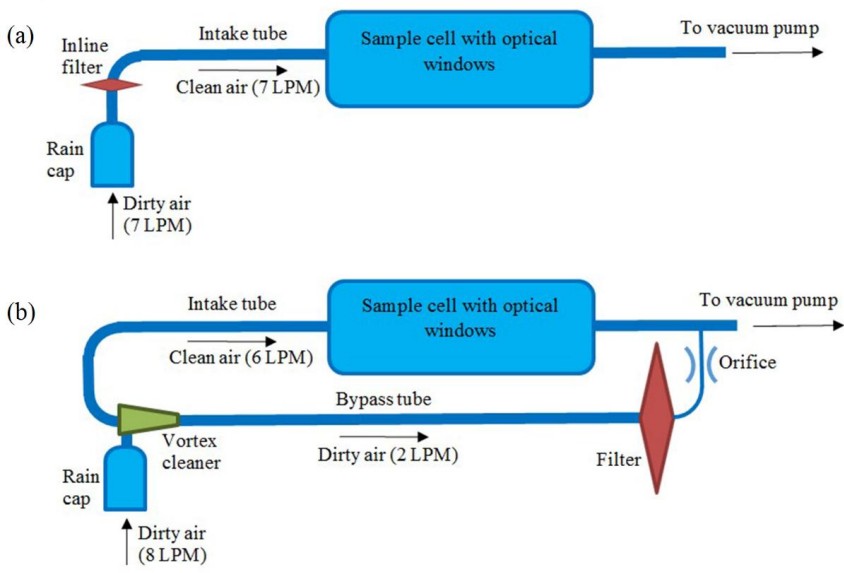


**Figure 3.** Schematics of EC155 sampling systems with (a) an original in-line filter and (b) vortex
intake.





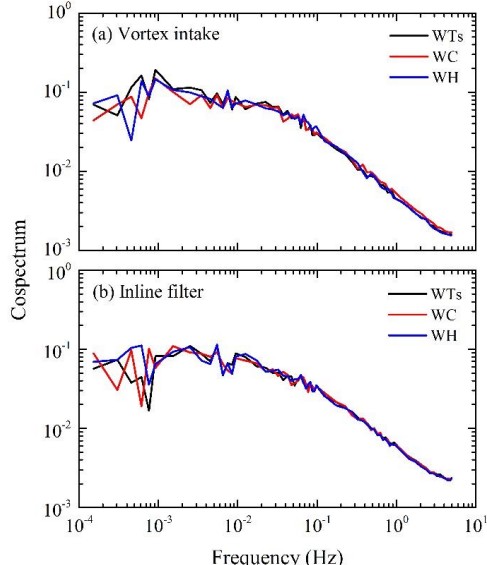


**Figure 4.** Cospectra of the EC-system (EC155) equipped with vortex intake (a) and in-line filter (b).

WTs, WC, and WH are cospectra of vertical wind velocity with sonic temperature, $CO_2$, and $H_2O$.

Cospectra are calculated from high-frequency data at 10 Hz obtained from the Olympic Park. Data

points in the figure are binned averages from means of one-hour period (12:00-13:00 Beijing

Standard Time) for each day in April 2014 for the in-line filter-based EC measurements and in

August 2015 for the vortex intake-based measurements.





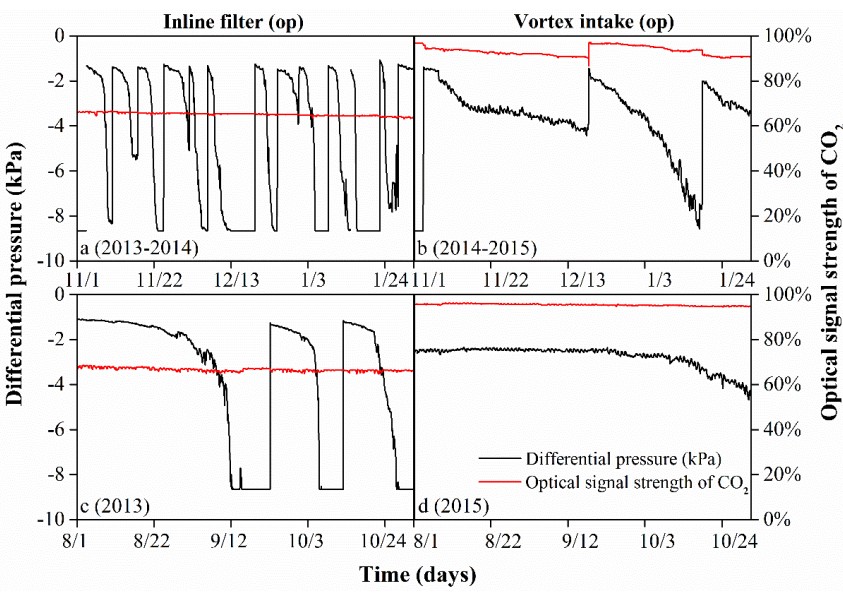


**Figure 5.** Differential pressure (black line) and optical signal strength of $CO_2$ (red line) of the EC-
system (EC155) equipped with vortex intake as compared to an in-line filter at the Olympic Park
(op); sub-figures (a) and (b) are for periods of very hazy conditions, whereas sub-figures (c) and (d)
are for periods of low haze.



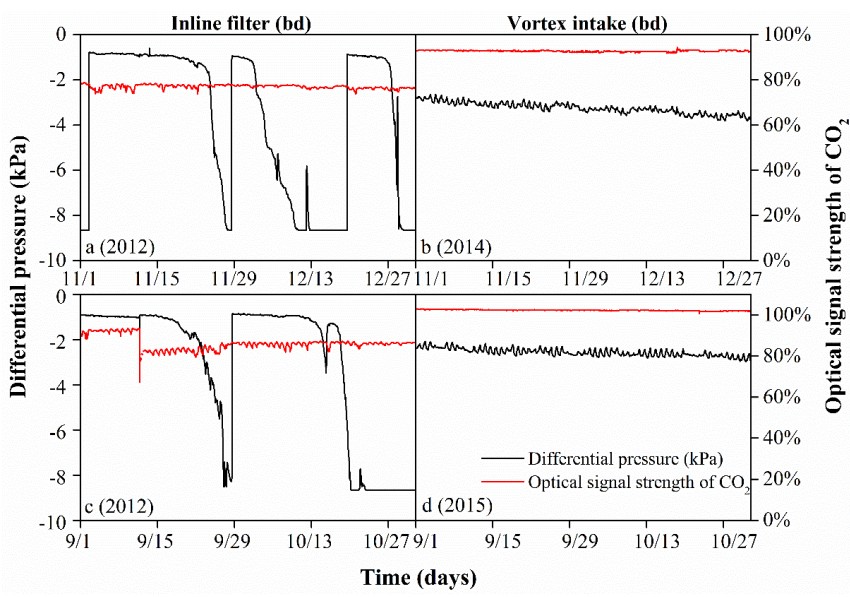


**Figure 6.** Differential pressure (black line) and optical signal strength of $CO_2$ (red line) of the EC-
system (EC155) equipped with vortex intake as compared to an in-line filter at the Badaling Farm
(bd); sub-figures (a) and (b) are for periods of very hazy conditions, whereas sub-figures (c) and (d)
are for periods of low haze.