# Peer review of "An innovative eddy-covariance system with vortex intake for measuring carbon dioxide and water fluxes of ecosystems"

_Atmospheric Measurement Techniques, 2016_

## Referee Comment (RC1) · Anonymous Referee #3 · 7 Nov 2016

Review of Manuscript "An innovative eddy-covariance system with vortex intake for 1 measuring carbon dioxide and water fluxes of ecosystems" by Ma et al., for AMT

This manuscript presents a new type of intake system for closed-path eddy covariance systems. The new, vortex-based system filters the incoming stream of air and significantly reduces the maintenance burden of filter exchange relative to traditional systems. The need for such a system is evident from the paper's site choice – polluted airs of the Beijing Metropolis. The manuscript is convincing and clear and relevant to the readership of Atmospheric Measurement Techniques.

The vortex inlet system is clearly described and has nice figures presenting its operation. When compared to a traditional in-line filter, both systems have co-spectra which

line up nearly perfectly with the temperature flux signal, so there is no apparent loss in data quality from the new system. I suggest the manuscript for acceptance following a few improvements, described below.

Major comments:

1. Please present the CO2 flux – both its calculation method and its values – in Figs 5-6 so we can see what range of flux conditions this method has been tested. This calculation / presentation is important to give context for Fig 4 as well and should present the separation distance from the sonic anemometer in each set-up from the closed-path inlet. The type of logger used and method of logging (e.g., voltage signals, SDM) should also be presented as this may affect high frequency attenuation.

2. Is the pattern shown in Fig 4 consistent for other months of the test periods? Particularly since the manuscript shows how variable the air quality conditions are (Fig 1, 5, 6), it will be useful to have cospectra presented from different months and periods of time. A maximum signal attenuation should then be presented.

Minor comments:

Line 93: consider adding references to other landscapes that may benefit from such a system – e.g., measurements during or near fire/burn events (forests, croplands, or prairies), measurements at remote sites with infrequent access where filter changing should be minimized, etc.

Line 207 – quantify how well this system performs, rather than the "Very well" given here.

Figs 5-6 – is it possible or relevant to add a timeline of the haze index presented in Fig 1 to these graphs? Could we then relate the haziness to the filter replacement time? And/or we could normalize the performance under similar conditions? Rather than a time series a specific quantified range could replace "periods of very hazy conditions" and "periods of low haze" in the figure caption.

[Figure]

Technical/editing suggestions:

Line 125: convert from "study site is" to "study sites are"

Line 137: consider whether "restoration" is a better word than "recovering"

Line 212 etc – I would tend to use a subscript for the "s" in "Ts"

Line 220 change the position of the apostrophe from the plural "s'" to the singular "'s" to follow "each"

Line 246 – probably "in" or "with" are better choices than "when the" for the size of frequencies

---

## Referee Comment (RC2) · M. Aubinet (Referee) · 23 Nov 2016

**General comment**

The new vortex system presented in this paper appears ingenious and original and has the potential to improve closed and enclosed eddy covariance systems. Its application field is large and would help field experimenters to spare maintenance time. The paper is concise, well structured, well written. I will not come back on comments of rev #3 with which I agree.

**Specific Comments**

In view of its principle, the system should be more efficient for heavier dust particles.

This paper shows that it works successfully in very polluted areas but it could be interesting to evaluate its performances in forest or grasslands ecosystems where dust is characterized by smaller particles.

Figure 4 suggests that the vortex system does not affect the system frequency response. However it has been tested in conditions that are not very challenging (as noted by the authours, the cospectra are already strongly attenuated above 0.1 Hz) and it could be interesting to test the system frequency response in conditions of higher frequency turbulence.

This two remarks could be added in the conclusion section and suggested as further research perspectives.

Miscellaneous

L25, L49 : The application field of the system is larger than announced by the authours as it may concern all trace gas analysers (not only $H_2O$ and $CO_2$) and also enclosed paths systems.

L192 (and below): the reference is in fact Aubinet et al (2000)

In reference list : check the order of the references (a.o. Burba is misplaced)

---

## Referee Comment (RC3) · Anonymous Referee #2 · 28 Nov 2016

Summary: ——

This paper describes a new intake (called the "vortex" intake) that is part of the Campbell Scientific EC155 eddy-covariance system. The subject matter is appropriate for the AMT journal and the topic is innovative and interesting. While I am completely convinced by the authors that the vortex inlet requires less maintenence than a standard in-line filter type system, which is especially important in "dirty" environments; I am less impressed with the instrument-comparison-setup and presentation of the limited results that are shown. Though the authors want the readers to believe the vortex inlet has similar or better frequency response, a more focussed field comparison should be presented to convince readers (or at least this reader).

[Figure]

General Comments: ————————

1. In the conclusions the authors state, "There was no signficant attenuation of high frequencies, compared to the in-line filter-based system." I have the following comments related to this statement:

- a much more robust field comparison would be to have both a vortex and standard inlet systems setup side-by-side and operated over a long time period. In that way, a true comparison between instrument can be performed and the relative frequency respone between sensors can be presented.

- This statment in the conclusions apparently comes from Fig. 4...there are several questions about this figure (listed below), but from what I understand these normalized cospectra are from mid-day...if you want to convince me that the two different inlet types agree, you need to show that they agree during more than just mid-day conditions—-some comparison over a range of stability conditions should be shown.

2. It is stated that the freq response of sonic temperature produces an "ideal" response (p. 11). The results in Fig. 4 appear to suggest that the $CO_2$ and $H_2O$ response are both exactly the same as that of temperature. One needs to keep in mind that this result is only for mid-day conditions and does not necessarily extend to other conditions (the discussion on p. 11-12 does not mention this important fact at all). I also don't see a "sharp attenuation when the frequencies > 0.1 Hz" for WC and WH...it looks to me that the WC, WH, and WT cospectra all follow each other very closely? Finally, why does the vortex inlet "warrant automating high-frequency spectral corrections"? Because of the comparison with WTs?

3. The vortex inlet appears to have a sharply angled side walls...was this element to the design given much consideration? There has been much research in the particle sampling community that might be relevant here..I give one example reference, but I'm sure there are many...

Hermann, M., Stratmann, F., Wilck, M., & Wiedensohler, A. (2001). Sampling characteristics of an aircraft-borne aerosol inlet system. Journal of Atmospheric and Oceanic Technology, 18(1), 7-19.

4. The fluid mechanics in the inlet look like they might be quite complicated.. Has any wind tunnel or CFD modeling of the inlet been made? How does it perform in high winds?

5. The english language usage could be improved in places (a few examples are below).

Specific Comments: ―――――――

* p.3, l.47, "extent" should be "extend"...

* p.4, l.85, isn't the system not working properly in the examples already given? (ie, what do you mean by extreme cases?)

* p.5, l.108, how often is "frequently"? every week? every month?

* p.6, l.113, "The vortex intake..."

* p.8, l.157, "vortex inlet", not "vortex design"

* p.8, l170-172 and Fig. 2..it would be nice to add arrows and label some of the components shown in Fig. 2 that are discussed within the text.

* p.9, l.180, details about the "large filter"?

* p.9, l.190, mounted on the tower at a height of 11.7 m. (similar fix to wording for the Badaling site).

* p.9, l.192, this statement is rather vaugue (or, is there a better reference with more specific details about these EC data?).

* p.10. l.207, what do you mean by "very well"? Can this result be quantified?

* p.17, "Mchale" should be "McHale" (similar for McPherson).

* p.25, Fig.4, these are normalized cospectra so please state that in the legend.

* p.26, Fig. 5, what happened to the vortex inlet that is shown in Fig 5b. It is clearly much better than the inline filter, but it also appears to have some problem. Was a specific component of the vortex system getting plugged up?

* p.27, Fig. 6, any idea about the cause of the small "wiggles" in the differential pressure time series for the vortex inlet (I don't see any such wiggles with the inline filter)?

* in both Figs 5 and 6 the filters are clearly getting quite dirty–but the optical signal strength (red line) for the inline filters is practically unchanged (even in the very dirty environments)...is there any explanation for this seemingly surprising result? Also, is there any reason why the optical signal strength for the inline filters shown all start so low (near 80%) while those for the vortex intakes are all close to 100%?

---

## Author Comment (AC1) · 11 Jan 2017

The comment was uploaded in the form of a supplement:
http://www.atmos-meas-tech-discuss.net/amt-2016-280/amt-2016-280-AC1-supplement.zip

---

## Author Comment (AC2) · 11 Jan 2017

The comment was uploaded in the form of a supplement:
http://www.atmos-meas-tech-discuss.net/amt-2016-280/amt-2016-280-AC2-supplement.zip
* * *

---

## Author Comment (AC3) · 11 Jan 2017

The comment was uploaded in the form of a supplement:
http://www.atmos-meas-tech-discuss.net/amt-2016-280/amt-2016-280-AC3-supplement.zip